# Successful Mechanical Thrombectomy for Basilar Artery Occlusion in a Pediatric Patient: A Case Report

**DOI:** 10.3390/biomedicines11102774

**Published:** 2023-10-13

**Authors:** Agnete Teivāne, Inga Naļivaiko, Kristaps Jurjāns, Jānis Vētra, Andris Veiss, Arina Novaša, Sarmīte Dzelzīte, Dainis Krieviņš, Evija Miglāne

**Affiliations:** 1Faculty of Residency, Riga Stradins University, LV-1007 Riga, Latvia; 2Neurology Department, Pauls Stradins Clinical University Hospital, LV-1002 Riga, Latvia; kristaps.jurjans@stradini.lv (K.J.); janis.vetra@stradini.lv (J.V.); arina.novasa@stradini.lv (A.N.); evija.miglane@stradini.lv (E.M.); 3Institute of Diagnostic Radiology, Pauls Stradins Clinical University Hospital, LV-1002 Riga, Latvia; inga.nalivaiko@stradini.lv (I.N.); andris.veiss@stradini.lv (A.V.); sarmite.dzelzite@stradini.lv (S.D.); 4Department of Neurology and Neurosurgery, Riga Stradins University, LV-1007 Riga, Latvia; 5The Red Cross Medical College, Riga Stradins University, LV-1007 Riga, Latvia; 6Department of Vascular Surgery, Pauls Stradins Clinical University Hospital, LV-1002 Riga, Latvia; dainis.krievins@stradini.lv

**Keywords:** pediatric, stroke, basilar artery occlusion, mechanical thrombectomy

## Abstract

Studies have shown the benefits of endovascular treatment (EVT) in adult stroke cases, but its application in pediatric stroke remains controversial. Despite evidence of improved outcomes in adults, there are no established recommendations for EVT in children. Conducting individual case reports and case series is vital to understanding its potential advantages and disadvantages in this context. In this case report, a 9-year-old male initially diagnosed with gastroenteritis developed sudden left-sided weakness 1 day after admission. Comprehensive imaging revealed acute ischemia in the cerebellum, indicating a basilar artery thrombus. Urgent endovascular treatment (EVT) was performed 8.5 h after the onset of neurological symptoms, achieving successful revascularization. The patient underwent rehabilitation and was later discharged with improved neurological status. Despite extensive investigations, the stroke’s origin remained unknown. After six months, the patient exhibited complete neurological recovery, highlighting the patient’s remarkable resilience.

## 1. Introduction

Stroke, commonly associated with the elderly, is not limited to a specific age group, as it can affect individuals of all ages, including infants and children. Within the pediatric population, the incidence rates of arterial ischemic stroke range from 1 to 2 per 100,000 children annually [1], with vertebrobasilar territory involvement observed in up to 36% of cases [2]. However, the incidence of isolated childhood basilar artery occlusion (BAO) and stroke remains poorly understood. Regardless of age, stroke imposes a substantial burden on morbidity and mortality. Nonetheless, the impact on pediatric patients can be particularly severe due to strokes occurring at a younger age, delayed diagnosis, and misdiagnosis, leading to potential long-term disabilities that persist throughout their lifetime. Up to 70% of survivors remain with neurological sequelae or epilepsy, resulting in significant socioeconomic consequences [3,4]. Understanding the unique aspects of pediatric stroke, including epidemiology, risk factors, diagnostic hurdles, and long-term effects, is crucial for developing effective prevention, early intervention, and rehabilitation strategies.

Although the efficacy of endovascular treatment (EVT) for large vessel occlusion stroke in adults is well established through numerous randomized clinical trials, the available data concerning EVT in pediatric patients remain limited. Nevertheless, noteworthy cases of successful EVT have been documented in children, including neonates as young as 14 h old [5]. Recent advancements in mechanical devices, encompassing both older and more contemporary models such as stent retrievers, have exhibited encouraging outcomes in adult patients. These developments provide a promising avenue for the potential extension of such treatments to the pediatric population.

This case report highlights a 9-year-old boy who suffered from BAO resulting in a stroke successfully managed with EVT. Through the examination of this specific case, we aim to make a valuable addition to the expanding body of evidence regarding the utility of EVT as a viable treatment approach in pediatric stroke cases.

## 2. Case Description

### 2.1. Case Presentation

A 9-year-old male was admitted to a regional hospital due to a persistent complaint of diarrhea, exceeding ten episodes in two days, accompanied by a single episode of vomiting, tension headache, and mild vertigo. Initial diagnostic assessments did not reveal any notable abnormalities, leading to a diagnosis of gastroenteritis. Following rehydration therapy, the patient’s overall condition improved, and discharge plans were being considered. However, on the subsequent day after admission, at approximately 6 pm, the patient suddenly experienced weakness localized to the left side of his body. Recognizing the urgency of the situation, an emergency consultation with a neurologist was promptly arranged. Two hours following the onset of these new symptoms, a computed tomography (CT) scan of the brain was conducted (Figure 1), which did not show any acute changes in the brain. Nevertheless, the CT scan did reveal two hypodense regions in the left cerebellum, which raised suspicion of malignancy, along with a hyperdense appearance of the basilar artery (BA). Additionally, a hypodense region was noted in the left thalamic region, suggesting a potential prior lacunar stroke.

Following 3.5 h and in consultation with a pediatric specialist, the decision was made to transfer the patient to the Children’s Clinical University Hospital for a more comprehensive diagnostic assessment. An emergency magnetic resonance imaging (MRI) was conducted (Figure 2), revealing acute ischemia within the vascularization area supplied by the left posterior inferior cerebellar artery (PICA). Moreover, this MRI also indicated involvement in regions supplied by the left anterior inferior cerebellar artery (AICA) and superior cerebellar artery (SCA). The magnetic resonance angiography (MRA) performed concurrently showed evidence of a thrombus causing occlusion in the distal one-third of the basilar artery (BA). This comprehensive imaging assessment illuminated the extent of the vascular compromise and provided critical information for further evaluation and treatment planning.

Although the MRI revealed significant brain damage, as evidenced by positive findings on FLAIR (fluid-attenuated inversion recovery) and DWI (diffusion-weighted imaging), and despite the elapsed time of approximately 8.5 h from the onset of symptoms, a collaborative decision involving a team of medical specialists including a pediatrician, radiologist, strokologist, and interventional radiologist was made to proceed with EVT as the patient was deemed ineligible for intravenous thrombolysis due to exceeding the time window. This decision was reached despite the limited experience with pediatric strokes, underlining the urgency of the situation.

Subsequently, three hours after the MRI, the patient was transferred to Pauls Stradins Clinical University Hospital, a comprehensive stroke center. Upon admission, the patient displayed a Glasgow Coma Scale (GCS) score of 15 but presented with left-sided hemiparesis, as assessed by the National Institutes of Health Stroke Scale (NIHSS), which recorded a score of 6. The left arm and leg exhibited no effort against gravity. Vital signs at admission were as follows: blood pressure at 106/75 mmHg, heart rate at 80 beats per minute, oxygen saturation (SpO_2_) at 100%, respiratory rate (RR) at 18 breaths per minute, and a body temperature of +37.6 °C. The patient was expeditiously transferred to the EVT suite for immediate intervention.

### 2.2. Interventional Procedure

Under the administration of general anesthesia, a retrograde ultrasound-guided puncture was performed in the right common femoral artery, utilizing a 5F sheath. Subsequent to this, a left vertebral angiogram was conducted, revealing a significant blockage in the upper third of the basilar artery. This obstruction resulted in an absence of contrast in both posterior cerebral arteries, which was graded as a thrombolysis in cerebral infarction (TICI) scale score of 0, indicating minimal to no blood flow.

To restore normal blood flow to the affected areas, an aspiration catheter known as “Sofia” (Terumo, Tokyo, Japan) was carefully positioned in the distal V2 segment. Throughout the procedure, multiple left vertebral arteriograms were performed, consistently illustrating blockages in the middle segment of the basilar artery. After six meticulous attempts, a final angiogram conclusively confirmed the complete removal of the obstructions in both the basilar artery and the posterior cerebral arteries, achieving a TICI scale score of 3, indicating full and successful revascularization (Figure 3).

Following the procedure, manual compression was applied to the puncture site, and the thrombectomy process, which lasted for a duration of 60 min, concluded without any complications or adverse events.

### 2.3. Outcome and Follow-Up

The day following EVT, an MRI of the brain was conducted (Figure 4), revealing hyperintense regions on the right side of the pons and the upper left cerebellum. Importantly, there were no indications of hemorrhagic imbibition, which was reassuring. In terms of treatment, the patient commenced a secondary stroke prevention regimen, consisting of Acetylsalicylic Acid at a daily dose of 75 mg. Additionally, low-weight molecular heparin 30 mg was initiated, administered twice daily to further mitigate the risk of complications. Just one day after the EVT procedure, the patient embarked on early rehabilitation, overseen by a team of physiotherapists and occupational therapists. Remarkably, the post-procedure period progressed without any significant complications, and the patient was transferred back to the Children’s Clinical University Hospital for continued evaluation and rehabilitation. At the time of transfer, the patient’s neurological status was characterized by plegia in the left arm and severe paresis in the left leg, with the latter exhibiting no voluntary effort against gravity. The patient’s GCS score was recorded as 14, and the NIHSS indicated a score of 8. The timeline of clinical and procedural data can be seen in Figure 5. 

The patient’s family history did not reveal any pertinent chronic illnesses, heart diseases, or strokes. Furthermore, extensive investigations aimed at uncovering the etiology of the stroke yielded no significant factors. These investigations encompassed thorough assessments, including examinations for thrombophilia gene mutations and vascular malformation panels, as well as evaluations of immunodeficiency, immunology, and infectious diseases. Holter monitoring, employed to detect heart rhythm irregularities, did not reveal any abnormalities. Additionally, transesophageal echocardiography ruled out the presence of interatrial shunts and the potential for a paradoxical embolism originating from the heart. In an effort to explore potential genetic factors, the patient was consulted by a geneticist, and a comprehensive exome analysis was conducted. Regrettably, this analysis did not identify any significant genetic mutations associated with the stroke. Despite the rigorous and meticulous investigative efforts, the precise origin of the stroke remains unknown. 

Throughout the patient’s hospitalization, a continuous rehabilitation regimen was implemented. After approximately three weeks, the patient was discharged and transferred to a specialized rehabilitation facility. At the time of discharge, there was notable improvement in the patient’s overall neurological status. The patient demonstrated limited arm movement against gravity and slight left-sided leg drift. Neurologically, the patient achieved a GCS score of 15, a NIHSS score of 3, and a mRS score of 3.

During the six-month follow-up, the patient exhibited a complete neurological recovery, as evidenced by an NIHSS score of 0 and an mRS score of 0. An MRI scan conducted at this time (Figure 6) revealed scarring with hypointense areas on the right pons in the FLAIR sequence. Importantly, magnetic resonance angiography (MRA) confirmed unobstructed flow in the basilar artery. This positive outcome marked a significant milestone in the patient’s journey toward recovery and underlined the remarkable resilience of the young patient in overcoming the challenges posed by this enigmatic stroke.

## 3. Discussion

Pediatric strokes due to arterial vessel occlusion are uncommon, occurring at a rate of 1 to 2 cases per 100,000 children annually [1], with BA involvement seen in 36% of cases [2]. These strokes can present with a wide range of neurological symptoms, which can be subtle and challenging to diagnose promptly, leading to significant delays in treatment [6]. As a result, morbidity and mortality rates are affected, and up to 70% of survivors face lasting neurological deficits, resulting in notable socioeconomic consequences [3,4].

In our patient, the collective symptoms emerged over a span exceeding 48 h before the diagnosis was established. However, these symptoms were not initially attributed to the onset of BAO due to their inconsistency with stroke indicators. Given that the precise etiological factor of the stroke remains unidentified, it prompts consideration as to whether the episode of acute gastroenteritis (marked by severe diarrhea) could have precipitated the development of BAO. However, this possibility was investigated and subsequently discounted, as the blood analysis indicated the absence of notable changes, thereby ruling out this pathogenic mechanism.

To delve into potential genetic influences, a geneticist consulted with the patient and carried out a whole-exome analysis. Unfortunately, this examination did not reveal any notable genetic mutations linked to the stroke. Whole-exome sequencing (WES) is often favored under constraints of time or resources; however, it is imperative to note that WES does not encompass the entirety of the genome. Rather, it exclusively targets exons or coding regions that constitute a mere 1–2% of the complete genome [7]. Consequently, disease-associated variations occurring in the unexamined exonic regions remain uncharted. Thus, considering the unidentified etiology, it is advisable to deliberate the potential application of whole-genome sequencing (WGS).

Effective treatments for the pediatric ischemic stroke population lack robust evidence from randomized clinical trials, with insights primarily drawn from isolated cases or case series. In the absence of such trials, consensus recommendations suggest that pediatric patients may receive intravenous tissue plasminogen activator (tPA) at an adult dose of 0.9 mg/kg. However, the application of intravenous tPA in pediatrics is limited by the narrow treatment window, typically within 4.5 h of symptom onset. Recent guidelines provide further guidance, recommending the consideration of intravenous thrombolysis for pediatric patients who present with persistent debilitating neurological deficits (defined as a NIHSS score ≥ 6) and confirmed major cerebral artery occlusion [8]. In our case, intravenous thrombolysis was not administered due to the extended time interval (approximately 5.5 h) between symptom onset and diagnosis, in addition to the presence of ischemic findings on a non-contrast CT scan.

Mechanical thrombectomy has advantages such as a longer treatment window and potentially lower bleeding risk compared to intravenous tPA. However, its efficacy in pediatric cases is still experimental due to the lack of conclusive clinical data. Many reported cases of pediatric acute ischemic stroke (AIS) treated with recanalization therapy show positive outcomes. However, many studies indicate that a significant proportion, ranging from one-third to one-half, of children with AIS who do not undergo intervention achieve a favorable outcome without functional deficits [9,10]. In cases of children with mild initial stroke severity scores, the potential risks of thrombectomy might outweigh the expected benefits. Pediatric-specific factors that require consideration before EVT involve diminished artery size (both in the groin and cerebral regions), weight-dependent restrictions for radiological contrast, and potential radiation exposure in young children [8].

In our specific case, the patient presented with an occlusion in the posterior circulation artery. Notably, the patient belonged to the school-age group, implying that their arteries are naturally smaller in diameter compared to those of adults. Despite this difference in size, the patient’s arteries were deemed adequate for the planned procedure. To ensure a cautious and precise approach, we opted to perform a femoral artery puncture guided by ultrasound. This technique was chosen to minimize the potential risk of vasospasm, a complication that can occasionally occur. In our scenario, we made a deliberate choice to exclusively employ an aspiration technique, foregoing the use of a stent retriever. We utilized an aspiration catheter named ‘Sofia’. This catheter features a remarkably soft distal tip, significantly reducing the risk of damaging blood vessels. Moreover, its exceptional flexibility renders it highly suitable for catheterization in smaller vessels. This decision was predicated on the consideration that introducing a stent retriever, being a foreign object, could potentially lead to further irritation of the artery and an increased likelihood of vasospasm. Following the completion of the procedure, we elected to employ manual compression at the puncture site to facilitate closure. This approach was chosen to ensure a secure and controlled closure of the arterial access point.

## Figures and Tables

**Figure 1 biomedicines-11-02774-f001:**
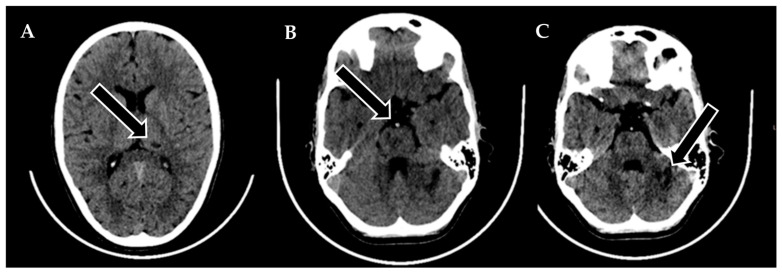
Emergency CT scan of the brain. (**A**) Hypodense region in the left thalamic region; (**B**) hyperdense basilar artery (HDBA) sign; (**C**) 2 hypodense regions in the left cerebellum.

**Figure 2 biomedicines-11-02774-f002:**
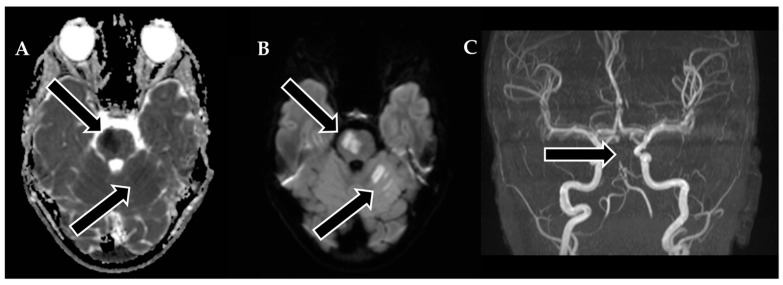
Emergency magnetic resonance imaging (MRI) before intervention. (**A**) Changes in the ADC map seen in the left PICA, as well as slightly in the AICA and SCA regions; (**B**) diffusion-weighted images showing acute changes in the left PICA, as well as slightly in the AICA and SCA regions; (**C**) MRA showing stop of flow in the distal 1/3 of the basilar artery.

**Figure 3 biomedicines-11-02774-f003:**
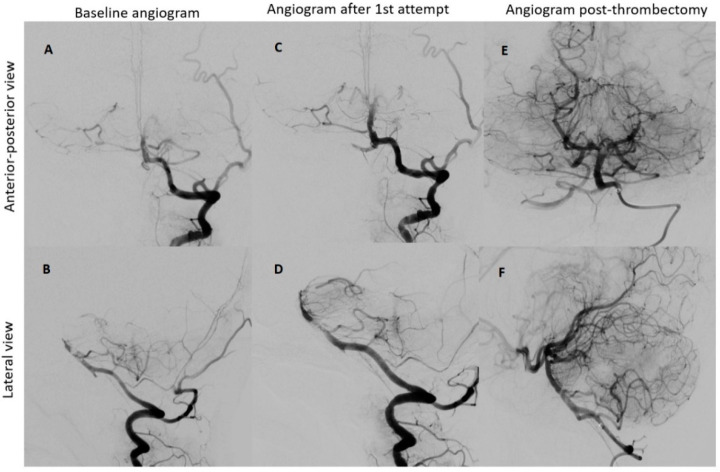
Neurointerventional procedure. (**A**,**B**) Initial angiogram from anterior–posterior and lateral views reveals an upper third basilar artery occlusion and no enhancement in both posterior cerebral arteries (TICI scale: 0). (**C**,**D**) After the first attempt, angiogram views show improved basilar artery enhancement, but filling defects and absent enhancement persist in both posterior cerebral arteries. (**E**,**F**) Post-thrombectomy angiograms demonstrate enhanced perfusion in the basilar artery and both posterior cerebral arteries (TICI scale: 3).

**Figure 4 biomedicines-11-02774-f004:**
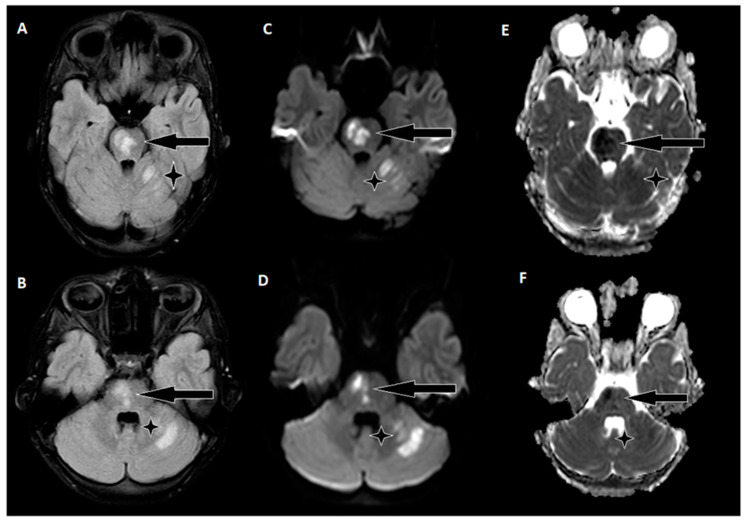
A follow-up brain MRI conducted two days after thrombectomy. (**A**,**B**) In the FLAIR sequence, hyperintense areas on the right side of the pons and upper left cerebellum. (**C**,**D**) Corresponding hyperintensities on diffusion-weighted images. (**E**,**F**) Hypointensities on the ADC map indicate acute ischemic stroke changes.

**Figure 5 biomedicines-11-02774-f005:**
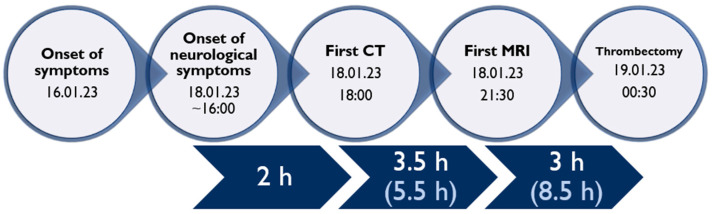
Timeline of clinical and procedural data.

**Figure 6 biomedicines-11-02774-f006:**
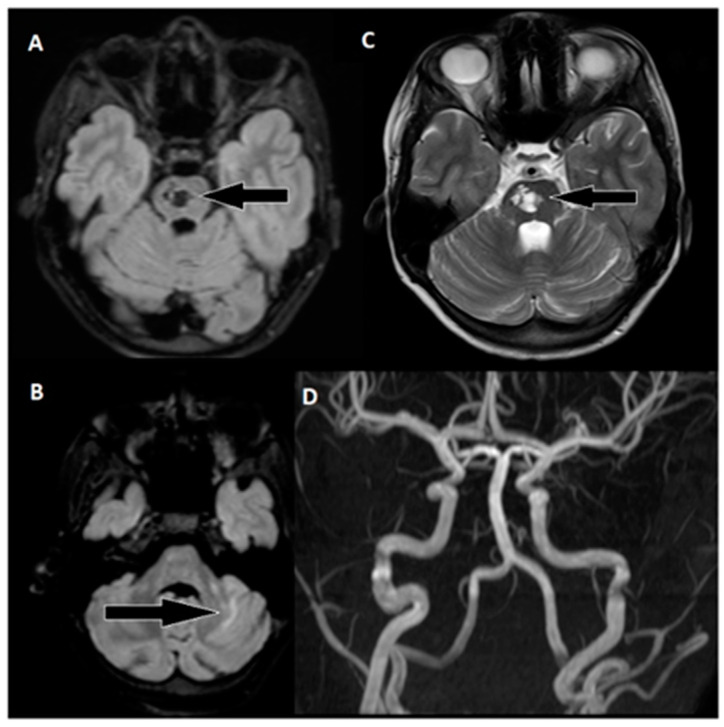
Follow-up MRI of the brain approximately 6 months after thrombectomy. (**A**) Markedly hypointense areas on the right side of the pons in the FLAIR sequence, indicating scarring. Corresponding hyperintense changes are observed on: (**C**) T2-weighted images, with a distinct hyperintense area on the right side of the pons (arrow). (**B**) Changes in the upper part of the left cerebellar hemisphere are visible in the FLAIR sequence. (**D**) An MRA examination shows unobstructed flow in the basilar artery and both cerebral posterior arteries.

## Data Availability

Not applicable.

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
