# Peer review of "Successful Mechanical Thrombectomy for Basilar Artery Occlusion in a Pediatric Patient: A Case Report"

_biomedicines, 2023, doi:10.3390/biomedicines11102774_

Round 1
Reviewer 1 Report
My congratulations to the operators for this highly challenging case. Following are my suggestion to improve the manuscript.
The Abstract section should be informative of the clinical case, since the reader often reads only this section, not the entire manuscript.
Line 39:LVO: you have never used this acronym later in the text. Therefore, you can delete it.
Lines 46-7: “…basilar artery occlusion (BAO)…”: replace with “…BAO…”.
Lines 47-8: “…endovascular treatment 47 (EVT).”: replace with “…EVT.”.
Line 58: “…16:00…”: replace with “…6 pm…”.
Lines 93-4: “…Endovascular Treatment (EVT)…”: replace with “…EVT…”.
Line 107: also add “ultrasound-guided” here, not only in the Discussion section.
Line 116: "Sofia": Manufacturer, City, State? Which are the features of this aspiration catheter? How does it work?
Line 135: which dose of low-weight molecular heparin was administerd?
Lines 98-101 vs 141-4:
“Upon admission, the patient displayed a Glasgow Coma Scale (GCS) score of 15 but presented with left-sided hemiparesis, as assessed by the National Institutes of Health Stroke Scale (NIHSS), which recorded a score of 6. The left arm and leg exhibited no effort against gravity.”
vs
“At the time of transfer, the patient's neurological status was characterized by plegia in the left arm and severe paresis in the left leg, with the latter exhibiting no voluntary effort against gravity. The patient's GCS score was recorded as 14, and the NIHSS indicated a score of 8.”
Based on GCS and NIHSS, it seems that the patient worsened after the procedure. If this is true, you should at least change the title of the manuscript in “Technically Successful Mechanical Thrombectomy for Basilar Artery Occlusion…”
Please, comment on that in the manuscript.
Line 187: “…basilar artery…”: replace with “…BA…”.
Lines 201-7: this paragraph must be introduced. I don’t understand the meaning of this concept in the whole context.
Line 208: I would specify again: change “…pediatric stroke population…” in “…pediatric ischemic stroke population…”.
Author Response
Thank You very much for the suggestions to improve the manuscript, we have taken them into account and made the following changes:
- The abstract has been rewritten to be more informative of the case.
- Line 39: LVO abbreviation - noted and removed.
- Lines 46-7: “…basilar artery occlusion (BAO) - noted and rewritten.
- Lines 47-8: “…endovascular treatment 47 (EVT) - noted and rewritten.
- Line 58: “…16:00…” - noted and rewritten.
- Lines 93-4: “…Endovascular Treatment (EVT)…” - noted and rewritten.
- Line 107: “ultrasound-guided” added.
- Line 116: "Sofia": noted and necessary information added in the case description as well as in the discussion part.
- Line 135: The dose was adjusted to the weight of the patients. The dosage was added to the text.
- Lines 98-101 vs 141-4: Certainly, I understand your concerns. After the procedure, there was a slight deterioration in the patient's condition. It's important to note that our facility primarily focuses on adult stroke cases, and while we facilitate EVT for pediatric patients, they are usually transferred back to the Children's University Hospital for continued care after a thorough evaluation, as was the case here. The changes observed in the GCS and NIHSS scores occurred a day apart, likely influenced by anesthesia, the post-operative period, and the progression of ischemic changes in the brain, rather than solely indicating the outcome of the case. Typically, we assess outcomes three months after the initial event, and in this instance, the results were positive with an NIHSS of 0 and an mRS of 0. Therefore, we believe it is unnecessary to modify the title of our manuscript.
- Line 187: “…basilar artery - noted and rewritten.
- Lines 201-7: Certainly, I understand your concerns. I apologize for not making it clear. Noted and rewritten.
- Line 208: Noted and rewritten.
We express our sincere gratitude again for your comprehensive review. Your insights have been invaluable in refining our manuscript. We believe that the revisions made bring us closer to the publication of this intriguing case report.
Reviewer 2 Report
Intersting case report. I have 2 questions. Did you think that the result would be avorable despite the long dealy to treatment and secondly was the patient treated with anticoagulants after discharge?
Author Response
Thank You for the revisions and questions regarding our case.
As a comprehensive stroke centre, our focus primarily lies outside of pediatric cases. Regrettably, we were informed about this particular case after a substantial delay. After a thorough assessment of the situation and careful consideration of the potential risks and benefits, we concluded that, given the typically grim outcomes associated with basilar artery occlusion and its high mortality rate, proceeding with the procedure seemed the most viable option. In this context, we felt that there was little to lose, hence our decision to move forward.
After discharge, the patient was prescribed antiplatelet therapy of 75 mg of Aspirin. This choice was made because no cardioembolic etiology was identified, aligning with the guidelines which clearly state that our selected therapy was appropriate in such cases.
Round 2
Reviewer 1 Report
Thank you for your comments to my revision of your manuscript.